# Dual-Function Metasurface for Tunable Selective Absorption

**DOI:** 10.3390/mi13122087

**Published:** 2022-11-26

**Authors:** Jingyu Zhang, Hanbing Yan, Xiaoqing Yang, Haohui Lyu

**Affiliations:** 1School of Electronics and Information Engineering, Sichuan University, Chengdu 610065, China; 2School of Control Engineering, Chengdu University of Information Technology, Chengdu 610225, China

**Keywords:** metasurface, tunable selective absorption, terahertz

## Abstract

Metamaterials have become a powerful technique in interdisciplinary research, especially in the field of designing terahertz devices. In this paper, two pairs of different structural units of aluminum–polymer composite metamaterials (APCM) for tunable selectivity are designed. One is designed to achieve high-contrast near-field imaging of linear polarized waves, the other is designed to achieve high-contrast near-field imaging of circularly polarized waves, which means the structural units have very large circular dichroisms. After theoretical design and simulation optimization, it can be found that the contrast of near-field imaging can be effectively controlled by using vanadium oxide (VO2) to fill the open gap of the structure. When the conductivity of VO2 is 200 S/m, both the reflection difference under linear polarization excitation and the reflection difference under the excitation of the circularly polarized wave are at the maximum. The former has a modulation depth of 0.8, and the latter has a modulation depth of 0.55. This work shows excellent tunable selective absorption ability, which will promote the application of metamaterials in terahertz absorber, such as biomedical, non-destructive testing, security inspection, wireless communication and so on.

## 1. Introduction

In recent years, the rapid development of terahertz technology has put forward higher requirements for terahertz devices. How to achieve flexible and efficient manipulation of terahertz waves has become one of the popular research topics. Metamaterial, known as a kind of artificial composite material with special properties, does not exist in nature. Through the special electromagnetic properties, metamaterials can break through the limitations of traditional materials. Their general properties can be changed by light and electromagnetic waves. In this process, many interesting experimental studies have been produced, such as negative refraction [1,2,3], perfect absorber [4,5,6], and near-field imaging [7,8,9,10]. Among them, using the characteristics of strong penetration and low energy, terahertz absorber technology has been widely used in many fields of electromagnetics, such as biomedical diagnosis [11,12,13], nondestructive testing [14,15], and safety inspection [16,17]. In general, the metamaterials for realizing the terahertz absorber consist of a sandwich structure, which usually consists of a top metal pattern, an intermediate insulating layer, and a bottom metal grounding layer. Its overall thickness is in the sub-wavelength scale. By exciting the different resonance modes of the metal surface and optimizing the thickness of the insulator layer, a high-contrast near-field imaging effect can be achieved. Polarization is an essential parameter of dealing with transverse waves [18]. Circular dichroism refers to the dichroism of circularly polarized light, that is, the differential absorption of left-handed circularly polarized light (LCP) and right-handed circularly polarized light (RCP). Unfortunately, the metasurface used for terahertz absorption is often polarization-sensitive [19,20,21,22,23], which is shown in Table 1. In addition, traditional metamaterials are also confronted with the problem of non-tunability. In view of the limitations of metamaterial structure, some dynamically adjustable special materials appear in our research. For example, vanadium oxide (VO_2_) can exhibit metal (insulating) properties at a relatively high temperature (relatively low temperature) [24]. By introducing these special materials into metamaterials, the contrast of near-field imaging can be freely controlled.

## 2. Design and Methods

In this paper, an aluminum–polymer composite metamaterial (APCM) for tunable selective absorption was designed, as shown in Figure 1a. In order to give the metasurface polarization-insensitive characteristics, two pairs of different structural units were designed. The boundary conditions were set to periodic unit cell boundaries during simulation. The reflection differences of different linearly polarized waves and different circularly polarized waves and the surface electric field distribution of the structural unit were calculated. Considering the contrast and imaging effect of near-field imaging, the influencing factors of ΔR_1_ and ΔR_2_ were studied, including the line width (W) of the aluminum pattern, the opening gap (G) of the aluminum pattern, and the height (H) of the polymide. In order to see the actual imaging effect, the near-field images of the metasurfaces at z = 220 μm were calculated. In addition, the opening gap of the designed structure was filled with vanadium oxide (VO_2_) material to dynamically control the contrast of near-field imaging. Additionally, the near-field imaging of two metasurfaces (10 × 10) at the conductivity of 200 S/m, 20,000 S/m, and 200,000 S/m were calculated and compared. All simulation results show beneficial effects on tunable selective absorption.

The designs and simulations were realized by using the Finite Element Method (FEM) (frequency domain solver in CST Studio Suite 2020 (Dassault Systemes Deutschland GmbH, France)).

The schematic diagram of two pairs of structural units for tunable selective absorption with different polarizations is shown in Figure 1, which is intuitively represented in the Cartesian coordinate system (x, y, z). Here, the yellow part represents aluminum, and the cyan part represents polymide. All structural units are sandwich structures composed of aluminum-polymer-aluminum patterns. The top layer is the aluminum pattern, the middle layer is the polymide, and the bottom layer is the metal aluminum sheet. The first pair of structural units was designed to achieve high-contrast near-field imaging of linear polarized waves. The second pair of structural units was designed to achieve high-contrast near-field imaging of circularly polarized waves, which means very large circular dichroism. The two pairs of structural units were complementary, respectively. We used the finite element method (FEM) to carry out a lot of simulation optimization, so the physical parameters of the structural unit are determined as shown below: W = 10 μm, G = 40 μm, B = 120 μm, H_m_ = 0.3 μm, H = 40 μm, P = 160 μm. The conductivity of aluminum is 3.72 × 10^7^ S*∕*m, the dielectric constant of the polymide is 3.5 and the loss tangent value is 0.0027. The VO_2_ follows the Drude model [25].
(1)$$\varepsilon \left(\omega \right)=\varepsilon_{\infty }-\frac{\omega_{p}^{2}\cdot \frac{\sigma }{\sigma_{0}}}{\omega^{2}+i\omega_{d}\omega }$$ where ω_d_ = 5.75 × 10^13^ s^−1^, ω_p_ = 1.40 × 10^15^ s^−1^, ε_ꝏ_ = 12 and σ_0_ = 3×10^5^ S/m. In the actual simulation, different permittivity and conductivity should be used for different phase states of VO_2_. Here, we can simulate the phase change of vanadium oxide (VO_2_) by changing the conductivity, and the range is from 200 S ∕ m to 200,000 S ∕ m. It should be noted that the boundary conditions along the x and y directions are all periodic boundaries. Since the skin depth of terahertz transmission waves on metal is about nanometers, we set the thickness of aluminum to 0.3 μm.

Next, we theoretically analyzed the working mechanism of the metasurface. According to the transmission theory of terahertz waves on metasurfaces, we can use Jones matrix to connect the reflected electric field with the incident electric field, which is expressed as [26]
(2)$$E_{i,r}(x,y,t)=\left(\begin{array}{c} E_{x}^{i,r}\\ E_{y}^{i,r} \end{array}\right)e^{i(kz-\omega t)}$$ where ω is the angular frequency, E_x_, E_y_, are complex amplitudes in the x and y directions, and k is the wave vector. The *I* and *r* represent incidence and reflection, respectively. The complex amplitudes of incident and reflection are related by the following relationship
(3)$$\left(\begin{array}{c} E_{x}^{r}\\ E_{y}^{r} \end{array}\right)=\left(\begin{array}{cc} r_{xx} & r_{xy}\\ r_{yx} & r_{yy} \end{array}\right)\left(\begin{array}{c} E_{x}^{i}\\ E_{y}^{i} \end{array}\right)$$ where *r_xx_*, *r_xy_*, *r_yx_* and *r_yy_* represent the reflection coefficients of the four components under the linear polarization base vector. Then, by combining the linear polarization basic vector and the circular polarization basic vector, the relationship between them can be obtained, which is described as follows:(4)$$\left(\begin{array}{cc} r_{RR} & r_{RL}\\ r_{LR} & r_{LL} \end{array}\right)=\frac{1}{2}\left(\begin{array}{cc} r_{xx}+r_{yy}+i(r_{xy}-r_{yx}) & r_{xx}-r_{yy}-i(r_{xy}+r_{yx})\\ r_{xx}-r_{yy}+i(r_{xy}+r_{yx}) & r_{xx}+r_{yy}-i(r_{xy}-r_{yx}) \end{array}\right)$$ where *r_RR_*, *r_RL_*, *r_LR_*_,_ and *r_LL_* represent the reflection coefficients of the four components under the circular polarization base vector. It is worth mentioning that *r_RR_* and *r_LL_* here stand for right-hand circularly polarized (RCP) waves and left-hand circularly polarized (LCP) waves, respectively.

## 3. Results and Discussions

First, we studied the reflection spectrum of two pairs of structural units under different polarized waves. As shown in Figure 2a,b, when the APCM is excited by a linearly polarized wave, the reflection difference between the x-polarized wave and the y-polarized wave is around 1.19 THz and can reach approximately 0.85. It can be seen that the reflectivity of linear cross-polarized waves is very low and can be ignored. Similarly, we calculated the reflection spectrum of the complementary structure in the first pair of structural units. It can be noticed that the reflection spectrum of the x-polarized wave and the y-polarized wave are completely opposite, and the reflection difference is about −0.85. In addition, when the APCM is excited by a circularly polarized wave, the reflection difference between the LCP waves and the RCP waves is around 1.395 THz and can reach about 0.8. Similarly, we also calculated the reflection spectrum of the complementary structure in the second pair of structural units. It can be found that the reflection spectra of LCP waves and RCP waves are completely opposite, and the reflection difference is about −0.8. In order to deduce the reason for the strong reflection difference, we calculated the surface electric field distributions of the two pairs of structural units at 1.19 THz and 1.395 THz (the lowest reflectivity), as shown in the inset in Figure 2. It can be found that both linearly polarized waves and circularly polarized waves produce strong resonance absorption on the surface of the structure, which is the most important reason for the low energy.

In order to obtain higher contrast in near-field imaging, we explore the influence of structural parameters on reflection differences under different polarized waves. Here, the reflection difference under linear polarization excitation is defined as ΔR_1_ = (E_x_ - E_y_) × 100%. The reflection difference under the excitation of the circularly polarized wave is defined as ΔR_2_ = (LCP - RCP) × 100%. Taking the APCM as an example, the influence of the line width (W) of the surface aluminum pattern on ΔR_1_ and ΔR_2_ are explored, as shown in Figure 3a. It can be found that when linearly polarized waves are incident, the peak value of ΔR_1_ first increases and then decreases with the increase of the (W). In this process, the reflection difference ΔR_1_ has a significant blue shift. Additionally, when W = 10 μm and W = 8 μm, the ΔR_1_ can be as high as about 0.9 and 0.85, respectively, which can meet our standards. Similarly, the peak value of ΔR_2_ first increases and then decreases as the W increases when the circularly polarized waves are incident. In this process, the reflection difference ΔR_2_ also has a significant blue shift. When *f* = 1.395 THz and W = 10 μm, the ΔR_2_ can reach a maximum value of about 0.8. After comprehensive consideration, we set the line width (W) to 10 μm in the next exploration. Figure 3b shows the influence of the opening gap (G) of the surface aluminum pattern on ΔR_1_ and ΔR_2_. When linearly polarized waves are incident, the peak value of ΔR_1_ first increases and then decreases with the increase of G. This process is accompanied by a blue shift. Additionally, when G = 40 μm, the maximum value is about 0.86. When circularly polarized waves are incident, the peak value of ΔR_2_ first increases from 0 and then decreases with the increase of G. When G = 40 μm, the ΔR_2_ can reach a maximum value of about 0.8. Thus, the G is set to 40 μm in the following exploration. Then, let us determine the influence of the height (H) of the polymide on ΔR_1_ and ΔR_2_. As shown in Figure 3c, it can be found that when the linearly polarized waves are incident, the peak value of ΔR_1_ first decreases and then increases to a maximum value of about 0.92 with the increase of H. Additionally, in this process, the peak of ΔR_1_ first causes a red shift and then a blue shift with the increase of H. When circularly polarized waves are incident, the ΔR_2_ only has a peak value of about 0.8 at H = 30 μm. So, we set the height (H) of the polymide to 30 μm.

Next, two metasurfaces for the near-field imaging of linearly polarized waves and circularly polarized waves were designed, as shown in Figure 4a,d. Taking into account the convenience of calculation, the two metasurfaces designed both contained 20 × 20 structural unit arrays. In order to better contrast the imaging effect, these two metasurfaces were divided into four parts, and each part contained 10 × 10 structural units. When linearly polarized waves were incident, the near-field images of the metasurfaces at z = 220 μm were calculated, as shown in Figure 4b,c. The bright and dark areas of the near-field image under the incidence of the E_x_ polarized waves are diagonally distributed. It is worth noting that the bright and dark areas of the near-field image under the incidence of the E_x_ polarized waves and the E_y_ polarized waves are just opposite. Similarly, when circularly polarized waves were incident, the near-field images of the metasurfaces at z = 220 μm were calculated, as shown in Figure 4e,f. The bright and dark areas of the near-field image are also diagonally distributed under the incidence of RCP waves. Under the incidence of LCP waves, the bright and dark areas of the near-field image are also just the opposite of the conditions under the incidence of RCP waves. All imaging effects are very good, in line with our design expectations.

Finally, in order to dynamically control the contrast of the near-field imaging, we filled the opening gap of the designed structure with VO_2_ material, as shown in the inset in Figure 5 (because the APCM is used as an example above, the complementary structure of the APCM is used here as an example). Since different conductivity of VO_2_ means different degree of phase transition, the ΔR_1_ and ΔR_2_ at the conductivity of 200 S/m, 5000 S/m, 10,000 S/m, 20,000 S/m, 50,000 S/m, 100,000 S/m and 200,000 were calculated, as shown in Figure 5. It can be found that as the conductivity increases, ΔR_1_ gradually decreases from the maximum value of 0.85. Additionally, the peak of ΔR_1_ has blue shift and broadening. Similarly, ΔR_2_ also gradually decreases as the conductivity increases. Additionally, the peak of ΔR_2_ also has blue shift and broadening. When the conductivity is 200 S/m, both ΔR_1_ and ΔR_2_ are at the maximum. The modulation range of ΔR_1_ is from 0.05 to 0.85 and has a modulation depth of 0.8. In addition, the modulation range of ΔR_2_ is from 0.05 to 0.6 and has a modulation depth of 0.55. The results are shown in Table 2.

In order to see the actual imaging effect of the composed metasurface and obtain a high-contrast effect, we chose to calculate the near-field imaging of two metasurfaces (10 × 10) at the conductivity of 200 S/m, 20,000 S/m, and 200,000 S/m, as shown in Figure 6a–l. It is worth noting that the near-field imaging here is simulated under Z = 220 μm. It can be seen that the filling of VO_2_ does not affect the imaging law of the metasurfaces. The imaging effect and contrast are the best when the conductivity is 200 S/m. Secondly, when the conductivity is 20,000 S/m, the metasurfaces follow the imaging law but the imaging effect and contrast are not good. However, when the conductivity is 200,000 S/m, almost no imaging law can be seen on the metasurface, and it appears dark in all areas.

## 4. Conclusions

In summary, we designed an aluminum polymer composite metamaterial (APCM) structure for tunable selective absorption. Two pairs of different structural units are designed. We observed the reflection spectra of each unit under different polarization waves. The results show that both by the LCP and the RCP waves the bright and dark areas of the near-field image are diagonally distributed by the externally polarized light. In order to explore its physical mechanism, the reflection differences of different linear and circular polarized waves and the surface electric field distribution of structural units were analyzed. Additionally, the imaging contrast including line width, opening gap, and polyamide height of aluminum die on imaging contrast were studied. In order to dynamically control the contrast of the near-field imaging, we filled the opening gap of the designed structure with VO_2_ material and studied the effect of phase transition characteristics of VO_2_ on imaging contrast. Compared with the previously reported absorbers, the proposed absorber has two advantages: First, and most importantly, the structural units have very large circular dichroisms. Secondly, the contrast of near-field imaging can be effectively controlled. This method provides a new idea for the fabrication and application of terahertz tunable selective devices, which has many promising applications in the terahertz range, and increases the possibility of practicality.

## Figures and Tables

**Figure 1 micromachines-13-02087-f001:**
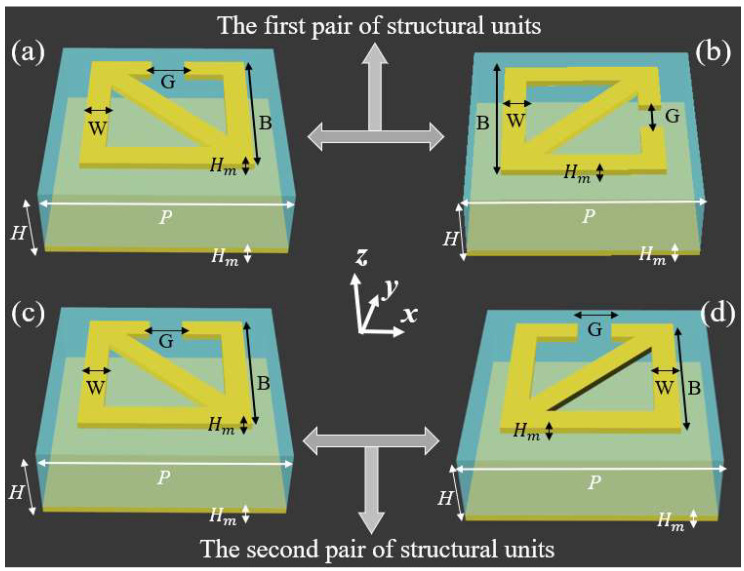
Schematic illustration of two pairs of structural units for tunable selective absorption of different polarized waves. (**a**) and (**b**) are the pair of structural units for tunable selective absorption of linear polarized waves. (**c**) and (**d**) are the pair of structural units for tunable selective absorption of circularly polarized waves.

**Figure 2 micromachines-13-02087-f002:**
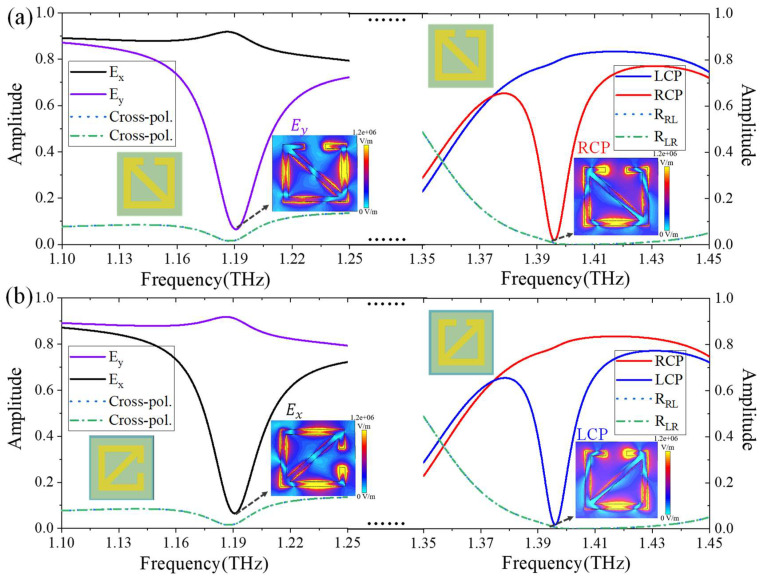
An illustration of the normalized reflection spectrum of different structural units excited by different polarized waves. (**a**) The reflection spectrum of the chiral structural units. (**b**) The reflection spectrum of the complementary structural units.

**Figure 3 micromachines-13-02087-f003:**
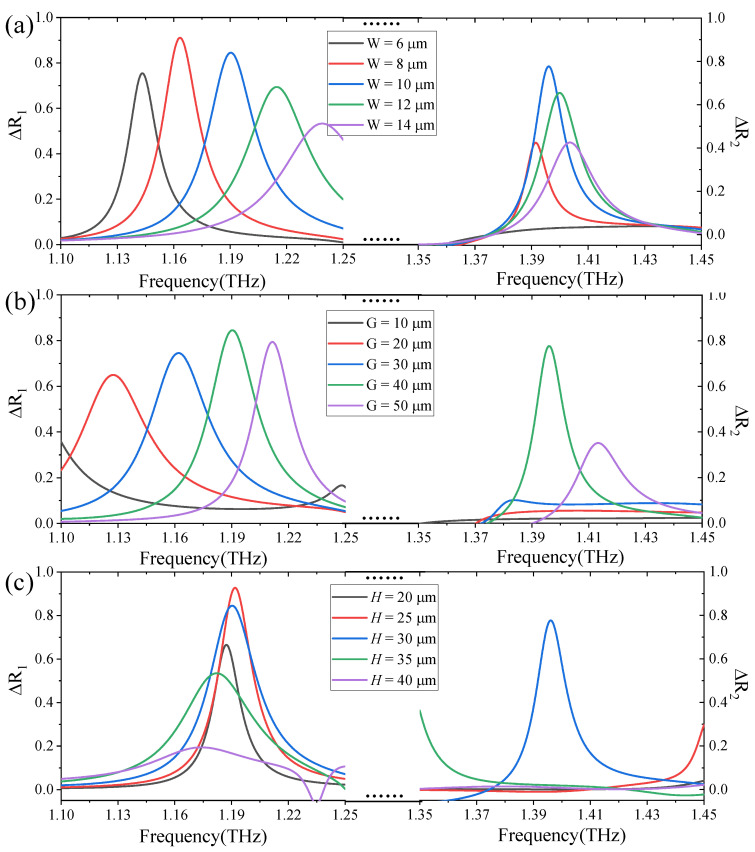
An illustration of the influence of the line width of the aluminum pattern, the opening gap of the aluminum pattern and the height of the polymide on ΔR_1_ and ΔR_2_. (**a**) Simulation results of ΔR_1_ and ΔR_2_ under different line width of the aluminum pattern. (**b**) Simulation results of ΔR_1_ and ΔR_2_ under different the opening gap of the aluminum pattern. (**c**) Simulation results of ΔR_1_ and ΔR_2_ under different the height of the polymide.

**Figure 4 micromachines-13-02087-f004:**
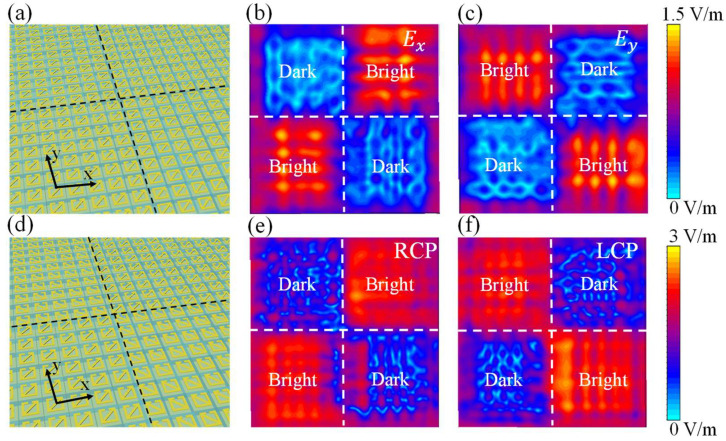
Simulation results of near-field imaging under different polarized waves. (**a**) Schematic diagram of metasurface for linearly polarized wave near-field imaging. (**b**) Simulation diagram of the actual effect of E_x_ polarized wave near-field imaging. (**c**) Simulation diagram of the actual effect of E_y_ polarized wave near-field imaging. (**d**) Schematic diagram of metasurface for circularly polarized wave near-field imaging. (**e**) Simulation diagram of the actual effect of RCP wave near-field imaging. (**f**) Simulation diagram of the actual effect of LCP wave near-field imaging.

**Figure 5 micromachines-13-02087-f005:**
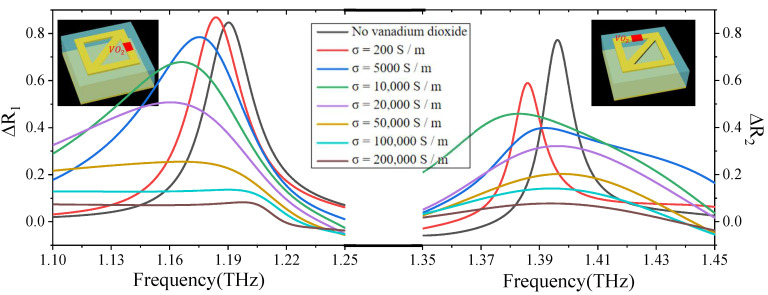
Simulation results of the influence of VO_2_ phase change on ΔR_1_ and ΔR_2_. The red part of the illustration represents VO_2_.

**Figure 6 micromachines-13-02087-f006:**
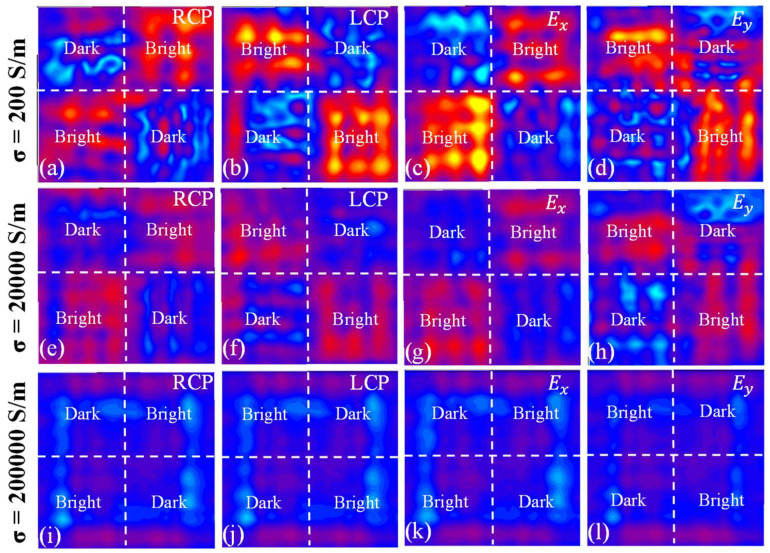
(**a**–**d**) are the actual effect diagrams of near-field imaging for linearly polarized waves and circularly polarized waves when the conductivity of vanadium dioxide is 200 S/m. (**e**–**h**) are the actual effect diagrams of near-field imaging for linearly polarized waves and circularly polarized waves when the conductivity of vanadium dioxide is 20,000 S/m. (**i**–**l**) are the actual effect diagrams of near-field imaging for linearly polarized waves and circularly polarized waves when the conductivity of vanadium dioxide is 200,000 S/m.

**Table 1 micromachines-13-02087-t001:** Comparison results of the reflection differences in this paper with previous similar works.

Linear Polarization/Circularly Polarization	ΔR_1max_ = (E_x_ − E_y_)_max_ × 100%	ΔR_2max_ = (LCP − RCP)_max_ × 100%	Frequency (THz)	Tunable	Ref.
Circularly polarization		Nearly 94%	5.6	NO	[19]
Circularly polarization		Nearly 65%	1.02	YES	[20]
Circularly polarization		Nearly 80%	0.486	YES	[21]
Linear polarization	Nearly 10%		1.1	NO	[22]
Linear polarization & Circularly polarization	Nearly 45%	Nearly 50%	2.0	NO	[23]
Linear polarization & Circularly polarization	Nearly 80%	Nearly 55%	1.19 & 1.395	YES	Our work

**Table 2 micromachines-13-02087-t002:** Simulation results of the influence of VO_2_ phase change on ΔR_1_ and ΔR_2._

Conductivity (S/m)	Frequency (THz)	ΔR_1_	Frequency (THz)	ΔR_2_
0	1.19	0.85	1.40	0.77
200	1.18	0.87	1.39	0.59
5000	1.18	0.79	1.39	0.40
10,000	1.17	0.68	1.38	0.46
20,000	1.16	0.51	1.40	0.33
50,000	1.17	0.26	1.40	0.21
100,000	1.19	0.14	1.40	0.14
200,000	1.20	0.09	1.40	0.08

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
