# Peer review of "Dual-Function Metasurface for Tunable Selective Absorption"

_micromachines, 2022, doi:10.3390/mi13122087_

Round 1

Reviewer 1 Report

Comments

1.     The introduction of the study is very short, and the literature review is also limitedly presented in the introduction section. A comparative table needs to be included for discussing the previous studies with the proposed study. Along with the literature review, the motivation, contribution, novelty, and organization of the study must be included in the introduction.

2.     In section 3, the results can be presented in tabular form and also briefly highlight the scientific findings.

3.     The novelty, originality, and future work of the study must be presented in the abstract and conclusion.

4.     The scientific findings of the study need to be presented in the abstract and conclusion section.

5.     In section 3, start the section with a paragraph rather than placing the figure. The same must be verified in the complete manuscript.

Reviewer 2 Report

This work designs an aluminum polymer composite metamaterial structure, which can selectively absorb linearly or circularly polarized light in terahertz band. Using this structure element, two structures are designed to achieve high contrast near-field imaging for linear and circular offset, respectively. In addition, by introducing vanadium dioxide and controlling its conductivity, the absorption spectrum can be adjusted. This has certain application potential in detection imaging.

Problems:

(1) This paper does not explain how to use this structure to achieve near-field imaging, which will make readers confused about the near-field imaging mentioned in the paper. If it is necessary to involve the content of near-field imaging, the authors should give the imaging system framework in the paper, and explain the imaging theory and process in detail. If it is not clear how to achieve near-field imaging with this structure, it is recommended to reduce the content about near-field imaging in this paper and focus on tunable selective absorbers in the terahertz band.

(2) Two structures are designed in this paper. The first one is used to deal with linear polarization in two orthogonal directions, and the second one is used to deal with left-handed and right-handed light. However, circularly polarized light will also have an impact on the first structure, and linear polarized light will also have an impact on the second structure. Does this have an impact on the overall detection results?

(3) It is suggested that this article quoteShen Z, He Q. Mutual circular polarization conversions in asymmetric transmission and reflection modes by three-layer metasurface with gold split-rings[J]. Optics Express, 2021, 29(21): 34850-34862.

(4) The logo on the authors should not use a and b, but use Arabic numerals as below.
